# A scalable platform for the development of cell-type-specific viral drivers

**Sinisa Hrvatin[1]\*, Christopher P Tzeng[1], M Aurel Nagy[1], Hume Stroud[1], Charalampia Koutsioumpa[1,2], Oren F Wilcox[1], Elena G Assad[1], Jonathan Green[1], Christopher D Harvey[1], Eric C Griffith[1], Michael E Greenberg[1]\***

[1]Department of Neurobiology, Harvard Medical School, Boston, United States; [2]BBS Program, Harvard Medical School, Boston, United States

**Abstract** Enhancers are the primary DNA regulatory elements that confer cell type specificity of gene expression. Recent studies characterizing individual enhancers have revealed their potential to direct heterologous gene expression in a highly cell-type-specific manner. However, it has not yet been possible to systematically identify and test the function of enhancers for each of the many cell types in an organism. We have developed PESCA, a scalable and generalizable method that leverages ATAC- and single-cell RNA-sequencing protocols, to characterize cell-type-specific enhancers that should enable genetic access and perturbation of gene function across mammalian cell types. Focusing on the highly heterogeneous mammalian cerebral cortex, we apply PESCA to find enhancers and generate viral reagents capable of accessing and manipulating a subset of somatostatin-expressing cortical interneurons with high specificity. This study demonstrates the utility of this platform for developing new cell-type-specific viral reagents, with significant implications for both basic and translational research.

DOI: https://doi.org/10.7554/eLife.48089.001

**\*For correspondence:**
shrvatin@gmail.com (SH);
Michael_Greenberg@hms.
harvard.edu (MEG)

**Competing interests:** The authors declare that no competing interests exist.

## Introduction

Enhancers are DNA elements that regulate gene expression to produce the unique complement of proteins necessary to establish a specialized function for each cell type in an organism. Large scale efforts to build a definitive catalog of cell types (*Cao et al., 2017*; *Rosenberg et al., 2018*; *Tasic et al., 2018*; *Zeisel et al., 2015*) based on their gene expression have recently successfully mapped epigenomic regulatory landscapes (*Buenrostro et al., 2015*; *Cusanovich et al., 2015*; *Mo et al., 2015*), enabling a mechanistic understanding of the underlying gene expression that is critical for cell-type-specific development, identity, and unique function. Importantly, characterization of individual enhancers has revealed their potential to direct highly cell-type-specific gene expression in both endogenous and heterologous contexts (*Dimidschstein et al., 2016*; *Graybuck et al., 2019*; *Jüttner et al., 2019*; *Mich et al., 2019*), making them ideal for developing tools to access, study, and manipulate virtually any mammalian cell type.

Despite recent success in cataloging the gene expression profiles of distinct cell subpopulations in the nervous system, our limited ability to specifically access these subpopulations hinders the study of their function. For example, the mammalian cerebral cortex is composed of over one hundred cell types, most of which cannot be individually accessed using existing tools. Glutamatergic excitatory neuron cell types propagate electrical signals across neural circuits, whereas GABAergic inhibitory interneuron cell types play an essential role in cortical signal processing by modulating neuronal activity, balancing excitability, and gating information (*Kepecs and Fishell, 2014*; *Marín, 2012*; *Rudy et al., 2011*). Although relatively lower in abundance than excitatory neurons, interneurons are highly diverse; for example, somatostatin-expressing cortical interneurons comprise several anatomically, electrophysiologically, and molecularly defined cell types whose dysfunction is

associated with neuropsychiatric and neurological disorders (*Jiang et al., 2015*; *Muñoz et al., 2017*; *Tasic et al., 2018*). Given the vast diversity of cell types in the brain, and the inability of our current tools to access most neuronal cell types, enhancer-driven viral reagents have the potential to become the next generation of cell-type-specific transgenic tools enabling facile, inexpensive, cross-species, and targeted observation and functional study of neuronal cell types and circuits.

Despite the potential of cell-type-specific enhancers to revolutionize neuroscience research, cell-type-restricted gene regulatory elements (GREs) have not yet been systematically identified. More-over, functional evaluation of candidate GRE-driven viral vector expression across all cell types in the tissue of interest is currently laborious, expensive, and low-throughput, typically relying on the pro-duction of individual viral vectors and the assessment of expression across a limited number of cell types by in situ hybridization or immunofluorescence. The lack of a generalizable platform for rapid identification and functional testing of cell-type-specific enhancers is therefore a critical bottleneck impeding the generation of new viral reagents required to elucidate the function of each cell type in a complex organism.

To address these issues, we merged the principles of massively parallel reporter assays (MPRA) (*Hartl et al., 2017*; *Inoue et al., 2017*; *Melnikov et al., 2012*; *Murtha et al., 2014*; *Patwardhan et al., 2012*; *Shen et al., 2016*) with single-cell RNA sequencing (scRNA-seq) (*Cao et al., 2017*; *Hrvatin et al., 2018*; *Klein et al., 2015*; *Macosko et al., 2015*; *Rosenberg et al., 2018*; *Stroud et al., 2017*; *Tasic et al., 2018*; *Tasic et al., 2016*; *Zeisel et al., 2015*), and developed a Paralleled Enhancer Single Cell Assay (PESCA) to identify and functionally assess the *specificity* of hundreds of GREs across the full complement of cell types present in the brain. In the PESCA proto-col, the expression of a barcoded pool of AAV vectors harboring GREs is analyzed by single-nucleus RNA sequencing (snRNA-seq) to evaluate the specificity of each constituent GRE across tens of thou-sands of individual cells in the target tissue, through the use of an orthogonal cell-indexed system of transcript barcoding (*Figure 1a*).

We validated the efficacy of PESCA in the murine primary visual cortex by identifying GREs that confine AAV expression to somatostatin (SST)-expressing interneurons and showed that these vec-tors can be used to modulate neuronal activity selectively in SST neurons. We chose to focus on SST neurons in the brain because this population is known to be diverse and to be composed of several relatively rare subpopulations (*Muñoz et al., 2017*; *Tasic et al., 2018*; *Tasic et al., 2016*), and thus might serve as a good test case. As described below, our findings highlight the utility of PESCA for identifying viral constructs that drive gene expression selectively in a subset of neurons and establish PESCA as a platform of broad interest to the research and gene therapy community, potentially enabling the generation of cell-type-specific AAVs for virtually any cell type.

## Results

### GRE selection and library construction

To identify candidate SST interneuron-restricted gene regulatory elements (GREs), we carried out comparative epigenetic profiling of the three largest classes of cortical interneurons: somatostatin (SST)-, vasoactive intestinal polypeptide (VIP)- and parvalbumin (PV)-expressing cells. To this end, we employed the recently developed Isolation of Nuclei Tagged in specific Cell Types (INTACT) (*Mo et al., 2015*) method to isolate purified chromatin from of each of these cell types from the cerebral cortex of adult (6–10 week-old) mice. The assay for transposase-accessible chromatin using sequencing (ATAC-Seq) (*Buenrostro et al., 2015*), which identifies nucleosome-depleted gene regu-latory regions, was then used to identify genomic regions with enhanced accessibility (i.e., peaks) in the SST (n = 57,932), PV (n = 61,108), and VIP (n = 79,124) chromatin samples (*Figure 1b,c*, *Fig-ure 1—figure supplement 1*, Materials and methods). These datasets can be used as a resource to identify putative gene regulatory elements as candidates for driving cell-type-specific gene expres-sion for the numerous subtypes of SST, PV or VIP-expressing intraneurons across diverse cortical regions.

To enrich for GREs that could be useful reagents to study and manipulate interneurons across mammalian species, including humans, we started with an expanded list of 323,369 genomic coordi-nates (*Supplementary file 1*) representing a union of cortical neuron ATAC-seq-accessible regions identified across dozens of experiments in our laboratory (Materials and methods, Stroud et al.,

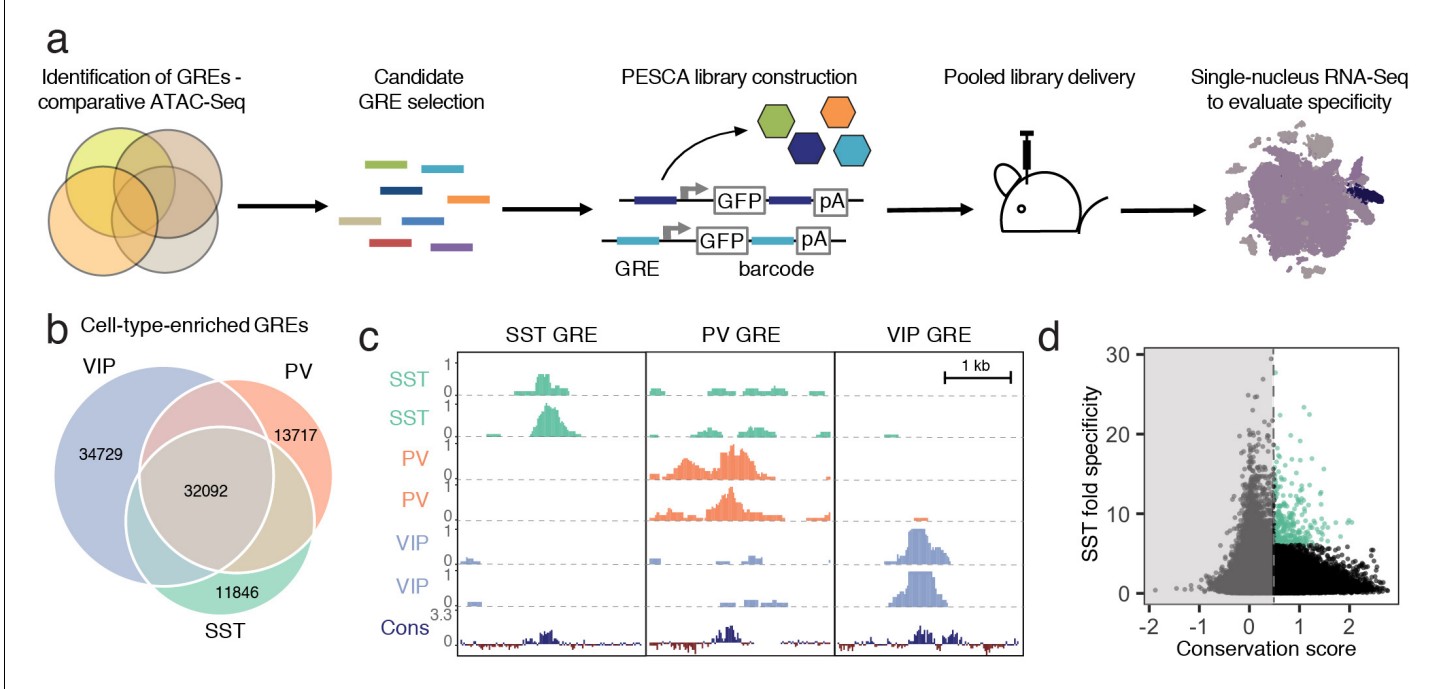

**Figure 1.** Experimental strategy and GRE selection. (**a**) Paralleled Enhancer Single Cell Assay (PESCA). Comparative ATAC-Seq is used to identify candidate GREs. A library of gene regulatory elements (GREs) is inserted upstream of a minimal promoter-driven GFP. The viral barcode sequence is inserted in the 3'UTR, and the vector packaged into rAAVs. Following *en masse* injection of the rAAV library, the specificity of the constituent GREs for various cell types in vivo is determined by single-nucleus RNA sequencing, measuring expression of the barcoded transcripts in tens of thousands of individual cells in the target tissue. Finally, bioinformatic analysis determines the most cell-type-specific barcode-associated rAAV-GRE-GFP constructs. pA = polyA tail. (**b**) Area-proportional Venn diagram of the number of putative GREs identified by ATAC-Seq of purified PV, SST, and VIP nuclei. Overlapping areas indicate shared putative GREs. Non-overlapping areas represent GREs that are unique to a single cell type. (**c**) Representative ATAC-seq genome browser traces of a putative GRE enriched in SST, PV, or VIP interneurons (normalized counts per location). Sequence conservation across the Placental mammalian clade is also shown. (**d**) Putative GREs (n = 323,369) are plotted based on average sequence conservation (phyloP, 60 placental mammals) and SST-specificity (ratio of the average ATAC-Seq signal intensity between SST samples and non-SST samples). Dashed vertical line indicates the minimal conservation value cutoff (0.5). Green coloring indicates the 287 most SST-specific GREs selected for PESCA screening.
DOI: https://doi.org/10.7554/eLife.48089.002

The following figure supplements are available for figure 1:

**Figure supplement 1.** Hierarchical clustering of the *Mo et al. (2015)* and our own ATAC-seq datasets.
DOI: https://doi.org/10.7554/eLife.48089.003
**Figure supplement 2.** Identification of conserved GREs.
DOI: https://doi.org/10.7554/eLife.48089.004

manuscript in preparation). We first filtered this initial set of 323,369 genomic coordinates to exclude GREs with poor mammalian sequence conservation (Materials and methods, *Supplementary file 1*, *Figure 1—figure supplement 2*). The remaining 36,215 genomic regions were ranked by an enrichment of ATAC-seq signal in the SST samples over PV/VIP (Materials and methods), and the top 287 most enriched GREs were selected for functional screening to identify enhancers that drive gene expression selectively in SST interneurons of the primary visual cortex (*Figure 1d*, *Supplementary file 2*).

A PCR-based strategy was used to simultaneously amplify and barcode each GRE from mouse genomic DNA (Materials and methods). To minimize sequencing bias due to the choice of barcode sequence, each GRE was paired with three unique barcode sequences. The resulting library of 861 GRE-barcode pairs was pooled and cloned into an AAV-based expression vector, with the GRE element inserted 5' to a promoter driving a GFP expression cassette and the GRE-paired barcode sequences inserted into the 3' untranslated region (UTR) of the GRE-driven transcript (Materials and methods, *Figure 2a*, *Figure 2—figure supplement 1*). This configuration was chosen to maximize the retrieval of the barcode sequence during single-cell RNA sequencing, which

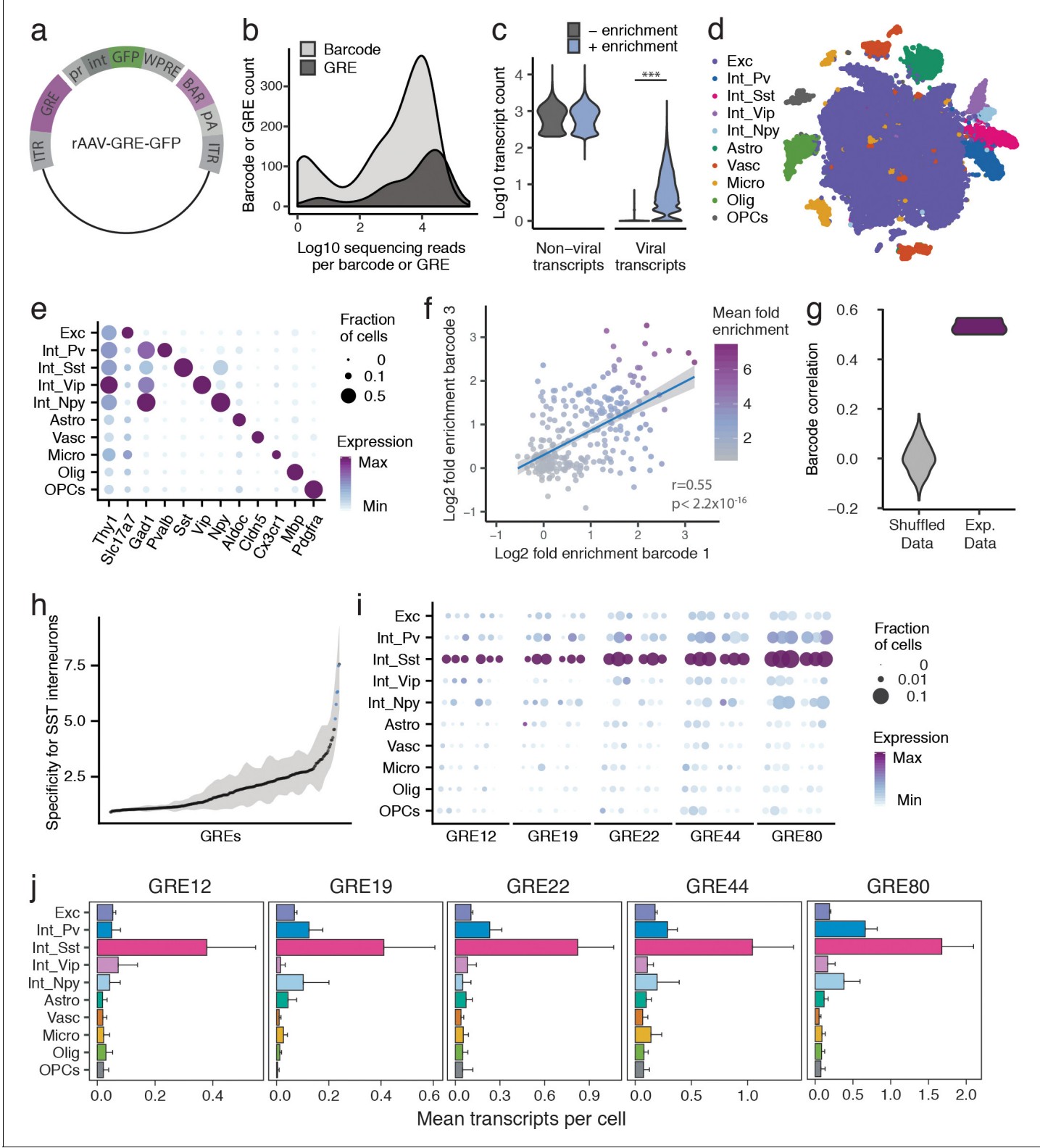

**Figure 2.** PESCA screen identifies GREs highly enriched for SST[+] interneurons. (**a**) PESCA library plasmid map. ITR, inverted terminal repeats; GRE, gene regulatory element; pr, HBB minimal promoter; int, intron; GFP, green fluorescent protein; WPRE, Woodchuck Hepatitis Virus post-transcriptional regulatory element; BAR, 10-mer sequence barcode associated with each GRE; pA, polyadenylation signal. (**b**) Library complexity plotted as distribution of the abundance of the 861 barcodes and 287 GREs in the AAV library. Barcodes and GREs were binned by number of sequencing reads attributed to

*Figure 2 continued on next page*

*Figure 2 continued*

each barcode or GRE within the library. (c) Transcript count per nucleus (n = 32,335 nuclei). Sequencing libraries were prepared with or without PCR-enrichment for viral transcripts. PCR enrichment resulted in a 382-fold increase in the number of recovered viral transcripts (p=0, Mann-Whitney U-test, two-sided) to an average of 15.6 unique viral transcripts per nucleus. Displayed as Log10(Count+1). (d) t-SNE plot of 32,335 nuclei from V1 cortex of two animals. Colors denote main cell types: Exc (Excitatory neurons), Pv (PV Interneurons), Sst (SST Interneurons), Vip (VIP interneurons), Npy (NPY Interneurons), Astro (Astrocytes), Vasc (Vascular-associated cells), Micro (Microglia), Olig (Oligodendrocytes), OPCs (Oligodendrocyte precursor cells). (e) Marker gene expression across cell types. Color denotes mean expression across all nuclei normalized to the highest mean across cell types. Size represents the fraction of nuclei in which the marker gene was detected. (f) Dot plot with each dot representing one GRE (n = 287). The values on each axis represent the Log2 SST fold-enrichment calculated for each GRE based on two of the three barcodes paired with that GRE - barcode one on the x-axis, and barcode three on the y-axis. Blue line indicates linear fit with 95% confidence intervals (shaded) (r = 0.55, $p<2.2 \times 10^{-16}$, Pearson's correlation). Color gradient indicates the average enrichment between the two barcodes. (g) Pairwise Pearson correlation between the enrichment values calculated from three sets of barcodes associated with 287 GREs for experimental data (Exp. Data, r = 0.52 ± 0.05, $p<2.2 \times 10^{-16}$, Pearson's correlation) and after random shuffling of enrichment values (Shuffled Data, r = 0 ± 0.06). (h) GREs ranked by average barcode expression specificity for SST interneurons across three barcodes. Shading indicates the minimal and maximal specificity calculated by analyzing each of the three barcodes associated with a GRE. Blue indicates the five top hits that also passed a statistical test for SST interneuron enrichment (FDR-corrected q < 0.01). (i) Expression of the top five hits: GRE12, GRE19, GRE22, GRE44, GRE80. For each GRE, expression values are split into two animals, and, for each animal, into the three barcodes associated with that GRE. Color denotes mean expression across all nuclei normalized to the highest mean across cell types. Size represents the fraction of nuclei in which the marker gene was detected. (j) Mean expression of GRE12, GRE19, GRE22, GRE44, and GRE80 across cell types. Error bars, s.e.m.

DOI: https://doi.org/10.7554/eLife.48089.005

The following figure supplements are available for figure 2:

**Figure supplement 1.** Schema for PESCA library construction.
DOI: https://doi.org/10.7554/eLife.48089.006

**Figure supplement 2.** UMI, gene and GRE detection metrics.
DOI: https://doi.org/10.7554/eLife.48089.007

**Figure supplement 3.** t-SNE plots of 32,335 nuclei from V1 cortex of two analyzed animals.
DOI: https://doi.org/10.7554/eLife.48089.008

**Figure supplement 4.** Pairwise comparison between SST fold-enrichment values.
DOI: https://doi.org/10.7554/eLife.48089.009

**Figure supplement 5.** GRE specificity metrics.
DOI: https://doi.org/10.7554/eLife.48089.010

**Figure supplement 6.** Analysis of computationally subsampled data.
DOI: https://doi.org/10.7554/eLife.48089.011

primarily captures the 3' end of transcripts. The human beta-globin promoter was chosen since it has previously been used in conjunction with an enhancer to drive strong and specific expression in cortical interneurons (*Dimidschstein et al., 2016*), although the modular cloning strategy is compatible with the use of other promoters. The library was packaged into AAV9, which exhibits broad neural tropism and has previously been used to drive payload expression in cortical neurons (*Cearley and Wolfe, 2006*). The complexity of the resulting rAAV-GRE library was then confirmed by next generation sequencing, detecting 802 of the 861 barcodes (93.1%), corresponding to 285 of the 287 GREs (99.3%) (*Figure 2b*).

## PESCA screen identifies GREs highly enriched for SST⁺ interneurons

To quantify the expression of each rAAV-GRE vector across the full complement of cell types in the mouse visual cortex, we used a modified single-nucleus RNA-Seq (snRNA-Seq) protocol to first determine the cellular identity of each nucleus and then quantify the abundance of the GRE-paired barcodes in the transcriptome of nuclei assigned to each cell type. Two adjacent injections (800 nL each) of the pooled AAV library ($1 \times 10^{13}$ viral genomes/mL) were first administered to the primary visual cortex (V1) of two 6-week-old C57BL/6 mice. Twelve days following injection, the injected cortical regions were dissected and processed to generate a suspension of nuclei for snRNA-Seq using the inDrops platform (*Klein et al., 2015*; *Zilionis et al., 2017*) (Materials and methods). A total of 32,335 nuclei were subsequently analyzed across the two animals, recovering an average of 866 unique non-viral transcripts per nucleus, representing 610 unique genes (*Figure 2—figure supplement 2a,b*).

Since droplet-based high-throughput snRNA-Seq samples the nuclear transcriptome with low sensitivity (*Klein et al., 2015*), viral-derived transcripts were initially detected in only 3.9% of sampled nuclei. We therefore designed a modified PCR-based approach to enrich for barcode-containing viral transcripts, which yielded deep coverage of AAV-derived transcripts with simultaneous shallow coverage of the non-viral transcriptome. PCR enrichment increased the viral transcript recovery 382-fold in the sampled nuclei, to an average of 15.6 unique viral transcripts, 6.0 unique GRE-barcodes, and 5.7 unique GREs per cell (*Figure 2b*, *Figure 2—figure supplement 2c*). Using this modified protocol, viral transcripts were identified across 86% of cells (*Figure 2—figure supplement 2d*), with a high correlation (r = 0.9, p<2.2 × 10$^{-16}$) observed between the abundance of each barcoded AAV in the library and the number of cells infected by that AAV (*Figure 2—figure supplement 2f*), suggesting that GRE sequences did not alter viral tropism and that GRE-driven vectors had broadly similar levels of expression. Only 0.3 ±0.06% (mean, stdev) of viral reads did not correspond to any of the known barcodes or could not be uniquely assigned to a barcode (within two mismatches), suggesting that this amplification strategy did not grossly change the composition of the viral library.

Nuclei were classified into ten cell types using graph-based clustering and expression of known marker genes (Materials and methods, *Figure 2c,d*, *Figure 2—figure supplement 3*). The average expression of each viral-derived barcoded transcript was analyzed across all ten cell types, and an enrichment score was calculated from the ratio of expression in *Sst*$^+$ nuclei compared to all *Sst*$^-$ nuclei. As expected, sets of three barcodes associated with the same GRE showed highly statistically correlated enrichment scores (r = 0.52 ± 0.05, p<2.2 × 10$^{-16}$) (*Figure 2e,f*, *Figure 2—figure supplement 4*), which were significantly lower when barcodes were randomly shuffled (shuffled r = 0.002 ± 0.06; Wilcox test between data and shuffled data, p=0.003).

Having confirmed a robust, non-random correlation in enrichment scores among the three barcodes associated with each GRE, we next computed a single expression value for each of the 287 viral drivers by aggregating expression data from three barcodes associated with the same GRE, and carried out differential gene expression analysis between *Sst*$^+$ and *Sst*$^-$ cells for each rAAV-GRE. Differential gene expression analysis between *Sst*$^+$ and *Sst*$^-$ cells for each rAAV-GRE revealed a marked overall enrichment of viral-derived transcripts in the *Sst*$^+$ subpopulation (*Figure 2—figure supplement 5a*). As expected, a high correlation was observed between GRE-specific enrichment scores across two animals (r = 0.54, p<2.2 × 10$^{-16}$) (*Figure 2—figure supplement 5b*). Among the 287 GREs tested, several viral drivers were identified that promoted highly specific reporter expression in the *Sst*$^+$ subpopulation (q < 0.01, fold-change >7, *Figure 2h–j*, *Figure 2—figure supplement 5c–e*). To assess how the abundance of each GRE in the library impacts our ability to detect cell-type-specific expression, we analyzed the specificity of each GRE as a function of the number of transcripts retrieved. We observed that highly abundant GRE-driven transcripts were more likely to be significantly enriched in SST$^+$ cells, suggesting that we may not have had sufficient power to assess the cell-type-specificity of the less abundant GREs in the library (*Figure 2—figure supplement 5f*). Consistent with this observation, computationally subsampling the number of viral transcripts across our most cell-type-specific GREs gradually reduced our ability to statistically detect their enrichment in *Sst*$^+$ cells (*Figure 2—figure supplement 6*). These observations suggest that the expression of sparsely detected GRE-driven transcripts may not be sufficient to allow evaluation of cell-type-specificity and that by increasing sequencing depth we may be able to screen and evaluate a larger number of GREs.

## In situ characterization of rAAV-GRE reporter expression

We next sought to validate the cell-type-specificity of the resulting hits using methods that do not rely on single-cell sequencing-based approaches. To this end, we selected three of the top five viral drivers (GRE12, GRE22, GRE44), as well as a control viral construct lacking the GRE element (ΔGRE), for injection into V1 of adult transgenic Sst-Cre; Ai14 mice, in which SST$^+$ cells express the red fluorescent marker tdTomato (*Supplementary file 3*). Fluorescence analysis twelve days following injection with rAAV-[GRE12, GRE22 or GRE44]-GFP revealed strong yet sparse GFP labeling centered around cortical layers IV and V (*Figure 3a–c*). By contrast, the control rAAV-ΔGRE-GFP showed a strikingly different pattern of GFP expression concentrated around the sites of injection, with expression in a larger number of cells (*Figure 3d*). Many rAAV-GRE12/22/44-GFP virally infected cells were SST-positive, as indicated by the high degree of overlapping GFP and tdTomato expression: 90.7 ± 2.1% for rAAV-GRE12-GFP (170 cells, four animals); 72.9 ± 4.2% for rAAV-GRE22-GFP (1164 cells,

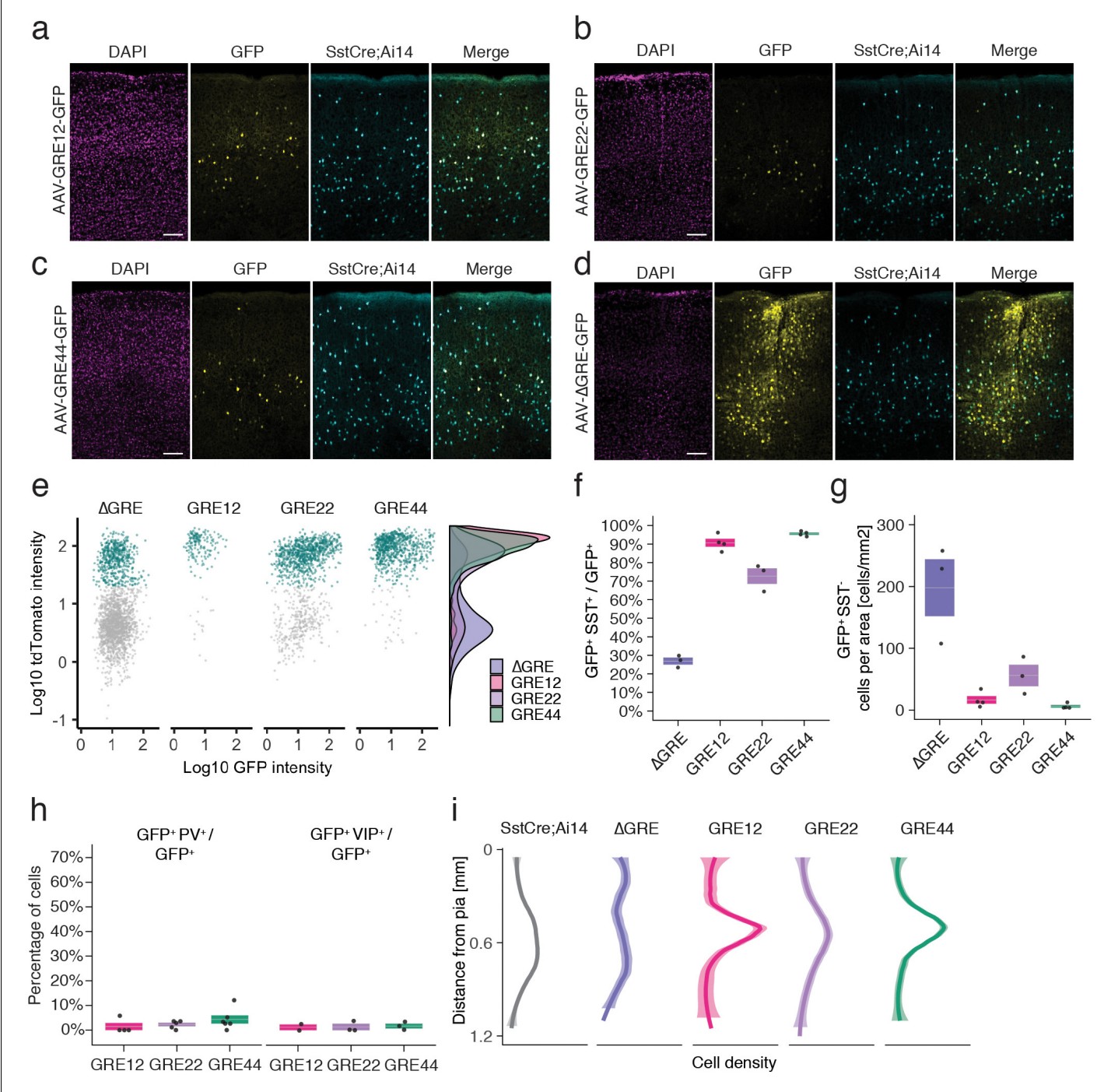

**Figure 3.** In situ characterization of rAAV-GRE reporter expression. (**a–d**) Fluorescent images from adult Sst-Cre; Ai14 mouse visual cortex twelve days following injection with rAAV-GRE-GFP as indicated. Scale bars 100 µm. (**e**) Identification of rAAV-GRE-GFP$^+$ cells that express tdTomato (SST$^+$). Each dot represents a GFP$^+$ cell (n = 2066, 172, 1164, and 765, for AAV-[ΔGRE, GRE12, GRE22, GRE44]-GFP, respectively). Cyan indicates tdTomato$^+$ (SST$^+$) cells. Distribution of cell frequency across tdTomato intensity is plotted on the right for each construct. (**f**) Quantification of the fraction of GFP$^+$ cells that are SST$^+$. Each dot represents one animal. Box plot represents mean ± standard error of the mean (s.e.m). Values are 27.2 ± 1.9%, 90.7 ± 2.1, 72.9 ± 4.2%, and 95.8 ± 0.6% for AAV-[ΔGRE, GRE12, GRE22, GRE44]-GFP, respectively. (**g**) Quantification of the number of GFP$^+$ SST$^-$ cells normalized for area of infection. Each dot represents one animal. Box plot represents mean ± standard error of the mean (s.e.m). Values are 198.0 ± 46.0, 16.4 ± 6.2, 56.0 ± 17.3 and 6.1 ± 2.1 cells/mm$^2$ for AAV-[ΔGRE, GRE12, GRE22, GRE44]-GFP, respectively. (**h**) Quantification of the fraction of GFP$^+$ cells that are PV$^+$ or VIP$^+$. Box plot represents mean ± standard error of the mean (s.e.m). Fraction of AAV-GRE-GFP$^+$ cells that are PV$^+$ is 1.4 ± 1.4%, 2.2 ± 0.7, and 4.3 ± 1.7% for AAV-[GRE12, GRE22, GRE44]-GFP, respectively. Similarly, the fraction of AAV-GRE-GFP$^+$ cells that are VIP$^+$ is 1.2 ± 1.2%, 1.3 ± 1.3%, and

*Figure 3 continued on next page*

*Figure 3 continued*

1.7 ± 1.0% for AAV-[GRE12, GRE22, GRE44]-GFP$^+$ cells, respectively. (i) Distribution of the location of GFP-expressing cells as function of distance from the pia. Gray represents SST$^+$ cells (n = 2648); Colored lines represents GFP$^+$ SST$^+$ cells (n = 2066, 172, 1164, and 765, respectively, for AAV-[ΔGRE, GRE12, GRE22, GRE44]-GFP). Shading represents the 95% confidence interval.

DOI: https://doi.org/10.7554/eLife.48089.012

The following figure supplements are available for figure 3:

**Figure supplement 1.** Fluorescent images from adult Sst-Cre; Ai14 mouse visual cortex twelve days following injection with rAAV-GRE-GFP as indicated.

DOI: https://doi.org/10.7554/eLife.48089.013

**Figure supplement 2.** Analysis of mDlx5/6-GFP$^+$ cells.

DOI: https://doi.org/10.7554/eLife.48089.014

**Figure supplement 3.** Quantification of the number of GFP$^+$ SST$^+$ cells normalized for area of infection.

DOI: https://doi.org/10.7554/eLife.48089.015

**Figure supplement 4.** Fluorescent images from adult Vip-Cre; Ai14 mouse visual cortex immunostained for PVALB twelve days following injection with rAAV-GRE-GFP as indicated.

DOI: https://doi.org/10.7554/eLife.48089.016

**Figure supplement 5.** Quantification of the fraction of GFP$^+$ cells that are present it each cortical layer.

DOI: https://doi.org/10.7554/eLife.48089.017

three animals), and 95.8 ± 0.6% for rAAV-GRE44-GFP (759 cells, four animals). (*Figure 3e,f*, *Figure 3—figure supplement 1*). By contrast, we observed that 27.2 ± 1.9% of GFP$^+$ cells also expressed tdTomato following rAAV-ΔGRE-GFP infection (2066 cells, three animals, *Figure 3e,f*). Although the 27.2% overlap between rAAV-ΔGRE-GFP expression and SST$^+$ cells suggests that our vector has some baseline preference for SST$^+$ interneurons, the insertion of GRE12, GRE22 and GRE44 serves to effectively restrict AAV payload expression to SST$^+$ interneurons. To show that our viral backbone could drive expression in non-SST cell types with the appropriate enhancer, we cloned the mDlx5/6 enhancer whose expression was restricted to a broader population of inhibitory neurons (*Dimidschstein et al., 2016*). We injected the rAAV2/9-mDlx5/6-GFP vector into Sst-Cre; Ai14 mice and observed that 57.1% of GFP$^+$ cells were not positive for tdTomato (1977 cells, three animals, *Figure 3—figure supplement 2*).

It is notable that the GREs seemingly not only promote expression in SST$^+$ cells but also greatly reduce background expression in SST$^-$ cells, indicating both enhancer and repressor functionality. Consistent with this hypothesis, the incorporation of GRE12, GRE22 and GRE44 into the rAAV both increased the number of SST$^+$ GFP$^+$ cells (1.7–2-fold) and dramatically (3–32-fold) decreased the number of SST$^-$ cells that expressed GFP (*Figure 3g*, *Figure 3—figure supplement 3*). To further investigate the specificity of our viral drivers among cortical interneuron cell types we injected each construct into Vip-Cre; Ai14$^+$ mice in which all VIP$^+$ cells express tdTomato, and used fluorescence antibody staining to label PV-expressing cells (*Figure 3—figure supplement 4*). Fluorescent signal analysis indicated the percentage of GFP$^+$ cells that were either VIP$^+$ or PV$^+$ (rAAV-SST12-GFP$^+$ [2.6 ± 2.6%], rAAV-GRE22-GFP$^+$ [3.5 ± 2.0%] and rAAV-GRE44-GFP$^+$ [6.0 ± 2.7%], *Figure 3h*). These findings confirm that among major interneuron cell classes, all three GRE-driven vectors are highly SST-specific.

Because at least five subtypes of cortical SST$^+$ interneurons have previously been identified based on the laminar distribution of their cell bodies and projections (*Muñoz et al., 2017*; *Urban-Ciecko and Barth, 2016*), we investigated the laminar distribution of GFP-expressing cells for the three SST-enriched viral drivers. Intriguingly, the majority of rAAV-GRE12-GFP$^+$ and rAAV-GRE44-GFP$^+$ SST$^+$ cells were found to reside in layers IV and V, which was distinct from the distribution observed for the full SST$^+$ cell population in visual cortex (p=1.3 × 10$^{-6}$, p<2.2 × 10$^{-16}$, respectively, Mann-Whitney *U* test, two-tailed, *Figure 3i*, *Figure 3—figure supplement 5*). By contrast, rAAV-ΔGRE-GFP was expressed in SST$^+$ cells as well as other neuronal subtypes across all layers, suggesting that increased labeling of rAAV-GRE12-GFP and rAAV-GRE44-GFP in layer IV and V was likely due to restricted gene expression and not restricted viral tropism.

## Electrophysiological characterization of rAAV-GRE-GFP-expressing SST subtypes

In addition to variability in laminar distribution, different electrophysiological phenotypes have also been observed in cortical SST interneurons (*Ma et al., 2006*; *Tremblay et al., 2016*). To determine whether AAV-GRE reporters can be used to distinguish electrophysiologically distinct SST subtypes, we injected our most cell-type-restricted construct, rAAV-GRE44-GFP, into the visual cortex of adult Sst-Cre; Ai14 mice and obtained whole-cell current-clamp recordings from double GFP- and tdTomato-positive neurons (rAAV-GRE44-GFP$^+$), as well as immediately nearby tdTomato-positive but GFP-negative cells (rAAV-GRE44-GFP$^-$).

Our recordings indicate that both rAAV-GRE44-GFP$^+$ and rAAV-GRE44-GFP$^-$ SST$^+$ neurons display the properties of adapting SST interneurons with high input resistances and features consistent with those previously reported for deep layer cortical SST neurons (*Ma et al., 2006*; *Xu et al., 2013*) (*Figure 4a,b*). However, rAAV-GRE44-GFP$^+$ SST neurons are distinct with respect to several electrophysiological parameters. The action potentials of rAAV-GRE44-GFP$^+$ SST neurons are significantly broader than those of rAAV-GRE44-GFP$^-$ SST neurons (*Figure 4c,d*), perhaps due to differences in expression of specific channels in these subgroups of SST neurons, such as voltage-activated potassium channels, and BK calcium-activated potassium channels (*Bean, 2007*; *Kimm et al., 2015*). Furthermore, rAAV-GRE44-GFP$^+$ SST neurons have a lower rheobase, and fire action potentials with a slower rising phase, and at lower maximal frequencies compared to rAAV-GRE44-GFP$^-$ SST neurons (*Figure 4a,d*, *Supplementary file 4*). Although we cannot confirm that GRE44 expression is restricted to a specific transcriptionally defined subtype of SST interneurons, our electrophysiology experiments further emphasize the potential of PESCA to target functionally distinct subgroups of previously defined interneuron types.

## Modulation of neuronal activity with rAAV-GREs

Finally, we evaluated whether the identified SST$^+$ neuron-restricted viral drivers support sufficiently high and persistent levels of payload expression to effectively modulate SST$^+$ cell physiology. Designer receptors exclusively activated by designer drugs (DREADDs) are a commonly employed viral payload used to dynamically regulate neuronal activity in response to the synthetic ligand clozapine-N-oxide (CNO) (*Armbruster et al., 2007*). We therefore injected the visual cortex of adult wild-type mice (6–8 week-old) with rAAV-GRE12-Gq-DREADD-tdTomato, a construct in which GRE12 drives the expression of an activating DREADD as well as tdTomato. GRE12 was chosen for this assay as it drives the weakest expression of the three evaluated GREs (*Figure 2e*) and thus, if it effectively drives DREADD expression, the other GREs might be expected to as well. We obtained electrophysiological recordings from tdTomato$^+$ cells of acute cortical slices in a whole-cell, current-clamp configuration two weeks post-injection. All tdTomato$^+$ cells showed striking sensitivity to CNO, as indicated by significantly increased firing rates in response to depolarizing current steps and depolarized resting membrane potentials (*Figure 4e–g*). To ensure that increases in firing rate upon CNO application were specific to infected SST$^+$ neurons, we obtained recordings from nearby uninfected pyramidal neurons that were identified by morphology and found that there was no statistically significant increase in firing rate upon CNO application (*Figure 4h–j*). These data demonstrate the ability of GRE-driven SST$^+$ neuron-specific reagents to robustly and specifically modulate the activity of SST$^+$ cells in non-transgenic animals.

## Discussion

The PESCA platform extends previous paralleled reporter assays (*Hartl et al., 2017*; *Inoue et al., 2017*; *Melnikov et al., 2012*; *Murtha et al., 2014*; *Patwardhan et al., 2012*; *Shen et al., 2016*) carried out using bulk tissue or sorted cells by including a single-cell RNA-seq-based readout (*Cao et al., 2017*; *Hrvatin et al., 2018*; *Klein et al., 2015*; *Macosko et al., 2015*; *Rosenberg et al., 2018*; *Stroud et al., 2017*; *Tasic et al., 2018*; *Tasic et al., 2016*; *Zeisel et al., 2015*) to evaluate the cell-type-specificity of gene expression. This represents a significant advancement over current approaches to viral vector design, as it enables the rapid in vivo screening of hundreds of GREs for enhanced cell-type-specificity without needing transgenic tools to evaluate their specificity. In this study, we applied PESCA to identify enhancer elements that robustly and specifically drive gene

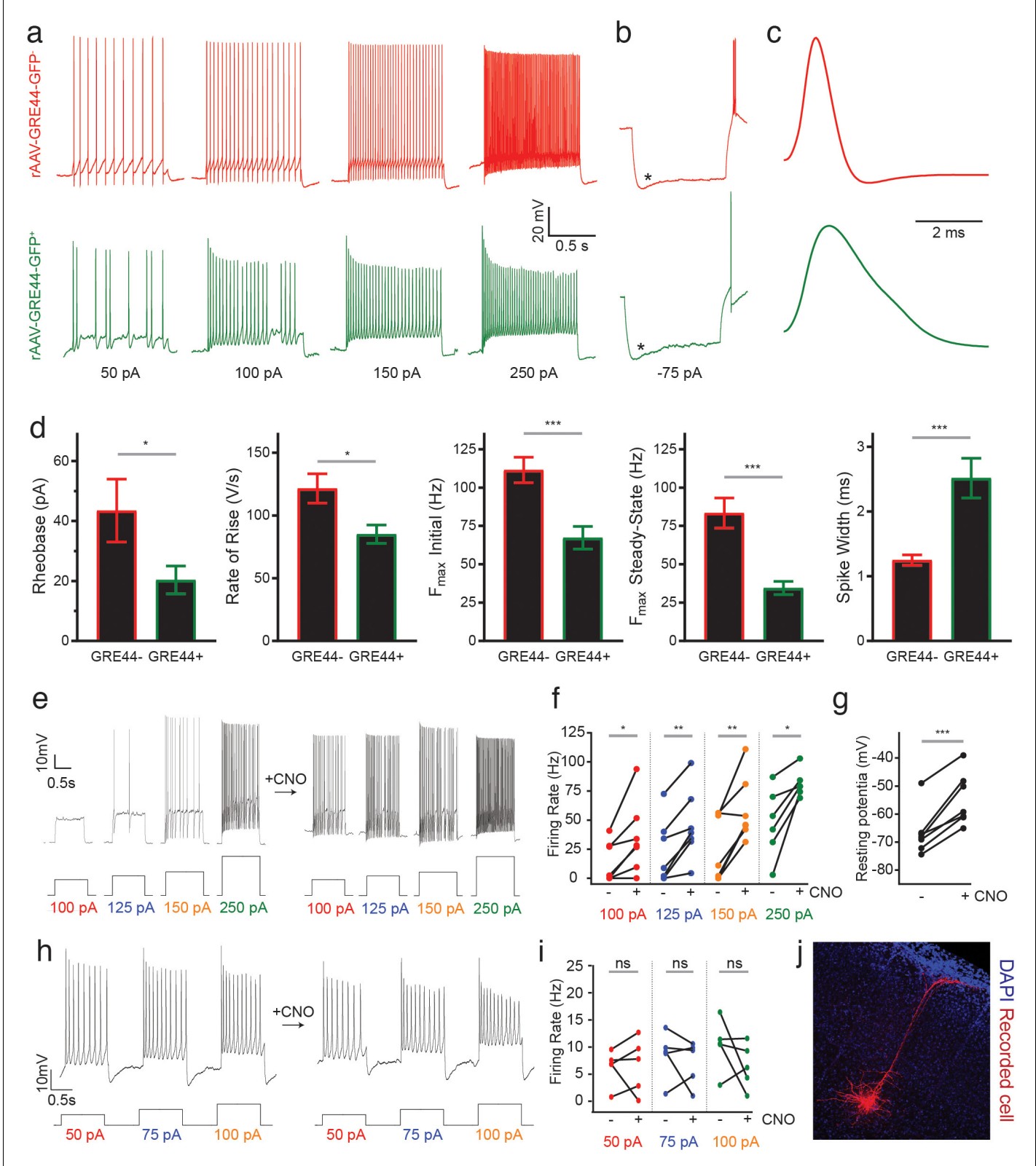

**Figure 4.** Electrophysiology of neurons expressing an rAAV-GRE-driven reporter and modulation of neuronal activity with rAAV-GREs. (a) Representative current-clamp recordings from SST neurons in the visual cortex of Sst-Cre; Ai14 mice injected with rAAV-GRE44-GFP. *Top*: Representative traces from a cortical SST neuron with Cre-dependent expression of tdTomato, in response to 1000 ms depolarizing current injections as indicated in black ('GRE44-'). *Bottom*: Traces from a tdTomato[+] SST neuron with GRE44-driven expression of GFP ('GRE44[+]'). GRE44- SST neurons
*Figure 4 continued on next page*

*Figure 4 continued*

were only recorded in the immediate vicinity of GRE44$^+$ SST neurons. (**b**) Recordings from GRE44$^+$ and GRE44$^-$ neurons in response to hyperpolarizing, 1000 ms currents. Asterisks indicate the sag likely due to the hyperpolarization-activated current $I_h$. Rebound action potentials following recovery from hyperpolarization, likely due to low-threshold calcium spikes mediated by T-type calcium channels, were also present in cells of both groups. Same scale as (**a**). (**c**) Broader action potentials in GRE44$^+$ SST neurons (bottom) compared to GRE44$^-$ SST neurons (top). Same vertical scale as (**a**) and (**b**). (**d**) Electrophysiological properties that differ between GRE44$^+$ (n = 16 cells from five mice) and GRE44$^-$ (n = 16 cells from four mice) SST neurons, including rheobase (minimal amount of current necessary to elicit a spike), maximal rate of rise during the depolarizing phase of the action potential, the initial and steady state firing frequencies (both measured at the maximal current step before spike inactivation), and spike width (measured as the width at half-maximal spike amplitude). *p<0.05; ***p<0.001, unpaired t-test, two-tailed. (**e**) Representative current-clamp recordings from AAV-GRE12-Gq-tdTomato$^+$ cells before and during CNO application. (**f**) Increased firing rates of AAV-GRE12-Gq-tdTomato$^+$ cells evoked by depolarizing current injections upon bath application of CNO (three animals, 6–7 cells). *p<0.05; **p<0.01, paired t-test, two-tailed. (**g**) Robust depolarization of AAV-GRE12-Gq-tdTomato$^+$ cells upon bath application of CNO (three animals, 6–7 cells). ***p<0.001, paired t-test, two-tailed. (**h**) Representative recordings from nearby uninfected pyramidal neurons in the visual cortex of mice that were injected with AAV-GRE-12-Gq-tdTomato$^+$, before (top) and during CNO application (bottom). (**i**) Firing rates of pyramidal neurons during CNO application remain unchanged (three animals, 5 cells). ns, p>0.05, paired t-test, two-tailed. (**j**) Representative image of a nearby recorded uninfected pyramidal neuron that was filled with neurobiotin.

DOI: https://doi.org/10.7554/eLife.48089.018

expression in a rare SST$^+$ population of GABAergic interneurons in the mouse central nervous system, although further work is needed to identify which specific molecular subtypes of SST interneurons are targeted. Since the vectors used in this PESCA screen in the absence of GREs show broad expression in the murine V1, the GREs we identified likely function to both enhance and restrict viral expression by a mechanism that remains to be explored.

In the future, several factors should be considered to facilitate the further optimization of the PESCA methodology for the development of cell-type-specific vectors. The selection of candidate GREs for screening will benefit from the systematic profiling of additional cell types by traditional or single-cell ATAC-Seq methods. In this regard, consideration of a published ATAC-Seq dataset from excitatory neurons (*Mo et al., 2015*) could have served to refine our starting GRE set by excluding approximately half of the screened GREs from our initial pool. This is particularly relevant insofar as the ability to assess the GRE library depends on the number of cells sequenced from the target and non-target populations and the sequencing depth, as the coverage of each GRE will be inversely proportional to the number of GREs screened. In the screen described here, we estimate having sufficient power to assess approximately 2/3 of the 287 GREs at the reported sequencing depth (*Figure 2—figure supplement 5*).

If a robust method of specifically isolating RNA from the target cell population is available, screening the PESCA library by sequencing pooled RNA from all target versus all non-target cells would provide a less expensive and potentially more scalable approach. However, by averaging across multiple non-target cell types, such an approach could be confounded by the presence of rare, highly expressing non-target cells.

Finally, once candidate PESCA hits have been identified, we suggest evaluating several follow-up assays at multiple titers to identify which among these hits have the desired intensity and specificity of protein expression. In this regard, the snRNA-seq PESCA screen identified GRE12, GRE22 and GRE44 as 8.3-, 9.1- and 7.2-fold more highly expressed in SST$^+$ compared to SST$^-$ cells, respectively, whereas these GREs showed distinct specificity for SST$^+$ cells (91%, 73% and 96% respectively, *Figure 3f*) when evaluated at the protein level, a finding which could be attributed to a variety of factors.

Given current evidence that the mechanisms of gene regulatory element function are conserved across tissues and species, it is likely that PESCA can be readily applied to other neuronal or non-neuronal cell types, diverse model organisms, tissues, and viral types. Moreover, single-cell screening approaches are not limited to GRE screening; PESCA can be easily adapted to assess the cell-type-specificity of viral capsid variants or other mutable aspects of viral design. Indeed, the PESCA library cloning strategy is largely vector- and capsid-independent, allowing for the use of different promoters or serotypes. Our choice of capsid and promoter was driven by previous work using AAV9 and the minimal beta-globin promoter to drive expression in cortical interneurons (*Dimidschstein et al., 2016*). However, different capsids or promoter may be preferred for targeting other cell types.

In conclusion, our study addresses the urgent practical need for new tools to access, study, and manipulate specific cell types across complex tissues, organ systems, and animal models by providing a screening platform that can be used to rapidly supply such tools as needed. Moreover, as the promise of gene therapy to treat and cure a broad range of diseases is being realized, PESCA has the potential to pave the way for a new generation of targeted gene therapy vehicles for diseases with cell-type-specific etiologies, such as congenital blindness, deafness, cystic fibrosis, and spinal muscular atrophy.

# Materials and methods

## Key resources table

| Reagent type (species) or resource | Designation | Source or reference | Identifiers | Additional information |
|---|---|---|---|---|
| Gene (*Mus musculus*) | *Sst* | | NCBI Gene ID: 20604 | |
| Genetic reagent (*M. musculus*) | Sst-IRES-Cre | Jackson Laboratory Stock # 013044 | IMSR Cat# JAX:013044, RRID:IMSR_JAX:013044 | |
| Genetic reagent (*M. musculus*) | Vip-IRES-Cre | The Jackson Laboratory Stock # 010908 | IMSR Cat# JAX:010908, RRID:IMSR_JAX:010908 | |
| Genetic reagent (*M. musculus*) | Pv-Cre | The Jackson Laboratory Stock # 017320 | IMSR Cat# JAX:017320, RRID:IMSR_JAX:017320 | |
| Genetic reagent (*M. musculus*) | SUN1-2xsfGFP-6xMYC | The Jackson Laboratory Stock # 021039 | IMSR Cat# JAX:021039, RRID:IMSR_JAX:021039 | |
| Genetic reagent (*M. musculus*) | Ai14 | The Jackson Laboratory Stock # 007914 | IMSR Cat# JAX:007914, RRID:IMSR_JAX:007914 | |
| Strain, strain background (*Escherichia coli*) | High Efficiency NEB 5-alpha | New England Biolabs | C2987H | Competent cells |
| Antibody | anti-GFP (Rabbit monoclonal) | Thermo Fisher | Cat# G10362; RRID:AB_2536526 | 0.012 ug/ul |
| Antibody | anti-Parvalbumin (Mouse monoclonal) | EMD Millipore | Cat# MAB1572; RRID:AB_2174013 | IF(1:2000) |
| Recombinant DNA reagent | pAAV-mDlx-GFP-Fishell-1 (plasmid) | PMID: 27798629 | Addgene # 83900; RRID:Addgene_83900 | |
| Recombinant DNA reagent | pAAV-ΔGRE -GFP- (plasmid) | This paper | | |
| Recombinant DNA reagent | pAAV-GRE12-GFP- (plasmid) | This paper | | |
| Recombinant DNA reagent | pAAV-GRE22-GFP- (plasmid) | This paper | | |
| Recombinant DNA reagent | pAAV-GRE44-GFP- (plasmid) | This paper | | |
| Commercial assay or kit | Nextera DNA Library Prep Kit | Illumina | FC-121–1030 | |
| Commercial assay or kit | In-Fusion HD cloning kit | Takara Bio | 639645 | |
| Commercial assay or kit | Agencourt AMPure XP | Beckman Coulter | # A63881 | |
| Commercial assay or kit | Hot Start High-Fidelity Q5 polymerase | New England Biolabs | M0494L | |

## Mice

Animal experiments were approved by the National Institute Health and Harvard Medical School Institutional Animal Care and Use Committee, following ethical guidelines described in the US National Institutes of Health Guide for the Care and Use of Laboratory Animals. For INTACT we crossed Sst-IRES-Cre (The Jackson Laboratory Stock # 013044), Vip-IRES-Cre (The Jackson Laboratory Stock # 010908) and Pv-Cre (The Jackson Laboratory Stock # 017320) with SUN1-2xsfGFP-6xMYC (The Jackson Laboratory Stock # 021039) and used adult (6–12 wk old) male and female F1 progeny. For PESCA screening we used adult (6–10 wk) C57BL/6J (The Jackson Laboratory, Stock # 000664) mice. For confirmation of hits we crossed Sst-IRES-Cre (The Jackson Laboratory Stock # 013044) and Vip-IRES-Cre (The Jackson Laboratory Stock # 031628) mice with Ai14 mice (The Jackson Laboratory Stock # 007914) and used adult (6–12 wk old) male and female F1 progeny. All mice were housed under a standard 12 hr light/dark cycle.

## INTACT purification and in vitro transposition

INTACT employs a transgenic mouse that expresses a cell-type-specific Cre and a Cre-dependent SUN1-2xsfGFP-6xMYC (SUN1-GFP) fusion protein. Nuclear purifications were performed from whole cortex of adult mice as previously described using anti-GFP antibodies (Fisher G10362) (*Mo et al., 2015*; *Stroud et al., 2017*). Isolated nuclei were gently resuspended in cold L1 buffer (50 mM Hepes pH 7.5, 140 mM NaCl, 1 mM EDTA, 1 mM EGTA, 0.25% Triton X-100, 0.5% NP40, 10% Glycerol, protease inhibitors), and pelleted at 800 g for 5 min at 4°C. DNA libraries were prepared from the nuclei using the Nextera DNA Library Prep Kit (Illumina) according to manufacturer's protocols. The final libraries were purified using the Qiagen MinElute kit (Cat# 28004) and sequenced on a Nextseq 500 benchtop DNA sequencer (Illumina). For each of the three inhibitory subtypes examined, we performed two independent ATAC-seq experiments, each on Sun1-positive nuclei isolated from a single animal. We did not count the nuclei prior to performing ATAC-seq, as yields were low enough that the process of counting would remove a large fraction of isolated nuclei and negatively impact the quality of the ATAC-seq experiment. However, during the process of establishing the Sun1 IP protocol, we consistently counted 20–30 k nuclei per animal.

## ATAC-seq mapping

All ATAC-seq libraries were sequenced on the Nextseq 500 benchtop DNA sequencer (Illumina). Seventy-five base pair (bp) single-end reads were obtained for all datasets. ATAC-seq experiments were sequenced to a minimum depth of 20 million (M) reads. Reads for all samples were aligned to the mouse genome (GRCm38/mm10, December 2011) using default parameters for the Subread (subread-1.4.6-p3) (*Liao et al., 2013*) alignment tool after quality trimming with Trimmomatic v0.33 (*Bolger et al., 2014*) with the following command: `java -jar trimmomatic-0.33.jar SE -threads 1 -phred33 [FASTQ_FILE] ILLUMINACLIP:[ADAPTER_FILE]:2:30:10 LEADING:5 TRAILING:5 SLIDINGWINDOW:4:20 MINLEN:45`. Nextera adapters were trimmed out for ATAC-seq data. Duplicates were removed with samtools rmdup. To generate UCSC genome browser tracks for ATAC-seq visualization, BEDtools was used to convert output bam files to BED format with the bedtools bamtobed command. Published mm10 blacklisted regions (*Schneider et al., 2017*) were filtered out using the following command: `bedops -not-element-of 1 [BLACKLIST_BED]`. Filtered BED files were scaled to 20 M reads and converted to coverageBED format using the `BEDtools genomecov` command. bedGraphToBigWig (UCSC-tools) was used to generate bigWIG files for the UCSC genome browser.

## ATAC-seq peak calling and quantification

Two independent peak calling algorithms were employed to ensure robust, reproducible peak calls. First, tag directories were created using HOMER makeTagDirectory for each replicate, and peaks were called using default parameters for findPeaks with -style factor. MACS2 was also called using default parameters on each replicate. The summit files output by MACS2 were converted to bed format and each summit extended bidirectionally to achieve a total length of 300 bp. As the ATAC-seq peak calls would ultimately be used to identify a small subset of highly enriched regulatory elements for subsequent screening, we required that a peak be called independently by both approaches in a

given replicate for its inclusion in the final peak list for that sample. This approach reduced the rate of false positive peak calls.

Beyond the ATAC-seq data generated for this manuscript (in SST, VIP, and PV populations), our laboratory has carried out several additional ATAC-seq experiments across cortical regions and cell types (DRD3, GPR26, NTSR1, SCNN1, CDH5, RBP4, RORB Cre driver x Sun1 crosses, manuscript in preparation). To produce a final list of reference coordinates containing 323,369 genomic regions that were accessible in at least one sample, the MACS2/HOMER-intersected peak bed files for each experimental replicate were unioned using the `bedops –everything` command. `Bedtools merge` was then used to combine any peaks that overlapped in this unioned bed file; in this way, any region that was significantly called a peak in at least one ATAC-seq dataset was incorporated in the final aggregated peak list of 323,369 neuronal ATAC-seq peaks. The featurecounts package was then used to obtain ATAC-seq read counts for each of these accessible putative GREs, for downstream enrichment analyses.

## Identification of conserved GREs

To identify GREs whose sequence is highly conserved across mammals, we first needed to identify an appropriate conservation score to use as a threshold for high conservation. We reasoned that by analyzing the conservation of DNA sequences of the same length, but an arbitrary distance of 100,000 bases away from each identified GRE, we would generate a set of DNA sequences whose conservation can be used to determine this threshold.

To this end, conservation scores for the 323,369 putative GREs and corresponding GRE-distal sequences were calculated using the `bigWigAverageOverBed` command to determine the average PhyloP score of each sequence based on mm10.60way.phyloP60wayPlacental.bw PhyloP scores (http://hgdownload.cse.ucsc.edu/goldenpath/mm10/phyloP60way/) (*Pollard et al., 2010*). After plotting the conservation score (phyloP, 60 placental mammals) of 323,369 GRE-distal sequences, we determined the conservation score of the 95th percentile of this distribution (PhyloP score = 0.5) and chose it as a minimal conservation score needed to classify any GRE as conserved. Using this cutoff, 36,215 GREs were classified as conserved and used for subsequent identification of SST-enriched GREs.

## Identification of SST-enriched GREs

We used genomic coordinates of 36,215 conserved GREs over which to quantify the ATAC-Seq signal from SST+, VIP+ and PV+ cells. A matrix was constructed representing the mean ATAC-Seq signal in SST+, VIP+ and PV+ cells for each of the 36,215 GREs and normalized such that the total ATAC-Seq signal from each cell population was scaled to $10^7$. Fold-enrichment was calculated for each region/GRE as [(Signal in cell type A)+0.5] / [mean(signal in cell types B and C)+0.5]. GREs were subsequently ranked based on fold-enrichment score.

## Viral barcode design

Viral barcode sequences were chosen to be at least three insertions, deletions, or substitutions apart from each other to minimize the effects of sequencing errors on the correct identification of each barcode. The R library 'DNAbarcodes' and following functions were used: `initialPool = create.dnabarcodes(10, dist = 3, heuristic='ashlock'); finalPool = create.dnabarcodes(10, pool = initialPool, metric='seqlev');`

The result was a list of 1164 10-base barcodes that fit our initial criteria.

## Amplification of GREs and barcoding

### Genomic PCR

PCR primers were designed using primer3 2.3.7 (*Untergasser et al., 2012*) such that a 150–400 bp flanking sequence was added to each side of the GRE. The forward primers contained a 5' overhang sequence for downstream in-Fusion (Clonetech) cloning into the AAV vector (5'-GCCGCACGCG TTTAAT). The reverse primers contained a 5' overhang sequence containing the recognition sites for AsiSI and SalI restriction enzymes (5'-GCGATCGCTTGTCGAC). Hot Start High-Fidelity Q5 polymerase (NEB) was used according to manufacturer's protocol with mouse genomic DNA as template.

### Barcoding PCR

The unpurified PCR products from the genomic PCR were used as templates for the barcoding PCR. A forward primer containing the sequence for downstream in-Fusion (Clonetech) cloning into the AAV vector (5'-CTGCGGCCGCACGCGTTTA) was used in all reactions. Reverse primers were constructed featuring (in the 5' → 3'direction): 1) a sequence for downstream in-Fusion (Clonetech) cloning into the AAV vector (5'-GCCGCTATCACAGATCTCTCGA), 2) a unique 10-base barcode sequence, and 3) sequence complementary with the AsiSI and SalI restriction enzyme recognition sites that were introduced during the first PCR (5'-GCGATCGCTTGTCGAC). Three different reverse primers were used for each of the GREs amplified during the genomic PCR. Hot Start High-Fidelity Q5 polymerase (NEB) was used according to the manufacturer's protocol.

## PESCA library cloning

All PCR reactions were pooled and the amplicons purified using Agencourt AMPure XP. The pAAV-mDlx-GFP-Fishell-1 was a gift from Gordon Fishell (Addgene plasmid # 83900). The plasmid was digested with PacI and XhoI, leaving the ITRs and the polyA sequence. in-Fusion was used to shuttle the pool of GRE PCR products into the vector. Following transformation into High Efficiency NEB 5-alpha Competent *E. coli* and recovery, SalI and AsiSI were used to linearize the AAV vector containing the GREs. The expression cassette containing the human HBB promoter and intron followed by GFP and WPRE was isolated by PCR amplification from pAAV-mDlx-GFP-Fishell-1. The expression cassette was ligated with the linearized GRE-library-containing vector using T4 ligase and transformed into High Efficiency NEB 5-alpha Competent E.coli to yield the final library. 50 colonies were Sanger sequenced to determine the correct pairing between GRE and barcode and the correct arrangement of the AAV vector.

## AAV preparation

The pooled PESCA library or individual AAV constructs (100 μg) were packed into AAV9 at the Boston Children's Hospital Viral Core. The titers (2–50 × 10$^{13}$ genome copies/mL) were determined by qPCR. Next generation sequencing using the NextSeq 500 platform was used to determine the complexity of the pooled PESCA library (*Figure 2a*).

## V1 cortex injections

Animals were anesthetized with isofluorane (1–3% in air) and placed on a stereotactic instrument (Kopf) with a 37°C heated pad. The PESCA library (AAV9, 1.9 × 10$^{13}$ genome copies/mL) was stereotactically injected in V1 (800 nL per site at 25 nL/min) using a sharp glass pipette (25–45 μm diameter) that was left in place for 5 min prior to and 10 min following injection to minimize backflow. Two injections were performed per animal at coordinates 3.0 and 3.7 mm posterior, 2.5 mm lateral relative to bregma, and 0.6 mm ventral relative to the brain surface.

Individual rAAV-GRE constructs were stereotactically injected at a titer of 1 × 10$^{11}$ genome copies/mL. (250 nL per site at 25 nL/min). All injections were performed at two depths (0.4 and 0.7 mm ventral relative to the brain surface) to achieve broader infection across cortical layers. The injection coordinates relative to bregma were 3.0 or 3.7 mm posterior, 2.5 or −2.5 mm lateral.

## Nuclear isolation

Single-nuclei suspensions were generated as described previously (*Mo et al., 2015*), with minor modifications. V1 was dissected and placed into a Dounce with homogenization buffer (0.25 M sucrose, 25 mM KCl, 5 mM MgCl$_2$, 20 mM Tricine-KOH, pH 7.8, 1 mM DTT, 0.15 mM spermine, 0.5 mM spermidine, protease inhibitors). The sample was homogenized using a tight pestle with 10 stokes. IGEPAL solution (5%, Sigma) was added to a final concentration of 0.32%, and five additional strokes were performed. The homogenate was filtered through a 40 μm filter, and OptiPrep (Sigma) added to a final concentration of 25% iodixanol. The sample was layered onto an iodixanol gradient and centrifuged at 10,000 g for 18 min as previously described[1,2](*Mo et al., 2015*; *Stroud et al., 2017*) . Nuclei were collected between the 30% and 40% iodixanol layers and diluted to 80,000–100,000 nuclei/mL for encapsulation. All buffers contained 0.15% RNasin Plus RNase Inhibitor (Promega) and 0.04% BSA.

## snRNA-Seq library preparation and sequencing

Single nuclei were captured and barcoded whole-transcriptome libraries prepared using the inDrops platform as previously described (*Klein et al., 2015*; *Zilionis et al., 2017*), collecting five libraries of approximately 3000 nuclei from each animal. Briefly, single nuclei along with single primer-carrying hydrogels were captured into droplets using a microfluidic platform. Each hydrogel carried oligodT primers with a unique cell-barcode. Nuclei were lysed and the cell-barcode containing primers released from the hydrogel, initiating reverse transcription and barcoding of all cDNA in each droplet. Next, the emulsions were broken and cDNA across ~3000 nuclei pooled into the same library. The cDNA was amplified by second strand synthesis and in vitro transcription, generating an amplified RNA intermediate which was fragmented and reverse transcribed into an amplified cDNA library.

For enrichment of virally-derived transcripts, a fraction (3 μL) of the amplified RNA intermediate was reverse transcribed with random hexamers without prior fragmentation. PCR was next used to amplify virally derived transcripts. The forward primer was designed to introduce the R1 sequence and anneal to a sequence uniquely present 5' of the viral-barcode sequence present in the viral transcripts (5'- GCATCGATACCGAGCGC). The reverse primer was designed to anneal to a sequence present 5' of the cell-barcode (5'- GGGTGTCGGGTGCAG). The result of the PCR is preferential amplification of the viral-derived transcripts, while simultaneously retaining the cell-barcode sequence necessary to assign each transcript to a particular cell/nucleus. Following PCR amplification (18 cycles, Hot Start High-Fidelity Q5 polymerase) all the libraries were indexed, pooled, and sequenced on a Nextseq 500 benchtop DNA sequencer (Illumina).

## inDrop sample mapping and viral barcode deconvolution by cell

The published inDrops mapping pipeline (github.com/indrops/indrops) was used to assign reads to cells. To map viral sequences, a custom annotated transcriptome was generated using the indrops pipieline's build_index command supplied with two custom reference files: 1. the GRCm38.dna_sm. primary_assembly.fa fasta genome with an additional contig for each viral barcode (comprising 5' sequence [gcatcgataccgagcgcgcgatcgc], barcode, and 3' sequence [tcgagagatctgtgatagcggc]) and 2. a GTF annotation file, with all viral sequences assigned the same gene_id and gene_name, but unique transcript_id, transcript_name, and protein_id. After inDrops pipeline mapping and cell deconvolution, the pysam package was used to extract the 'XB' and 'XU' tags, which contain cell barcode and UMI sequences, respectively, from every read that mapped uniquely to any one of the custom viral contigs (i.e. requiring the read map to the 10 bp barcode with at most one mismatch) in the inDrops pipeline-output bam files. These barcode-UMI combinations were condensed to generate a final cell x GRE barcode UMI counts table for each sample.

## Embedding and identification of cell types

Data from all nuclei (two animals, 5 libraries of ~3000 nuclei per animal) were analyzed simultaneously. Viral-derived sequences were removed for the purposes of embedding clustering and cell type identification. The initial dataset contained 32,335 nuclei, with more than 200 unique non-viral transcripts (UMIs) assigned to each nucleus. We recovered an average of 866 unique non-viral transcripts per nucleus, representing 610 unique genes. The R software package Seurat (*Butler et al., 2018*; *Satija et al., 2015*) was used to cluster cells. First, the data were log-normalized and scaled to 10,000 transcripts per cell. Variable genes were identified using the `FindVariableGenes()` function. The following parameters were used to set the minimum and maximum average expression and the minimum dispersion: x.low.cutoff = 0.0125, x.high.cutoff = 3, y.cutoff = 0.5. Next, the data was scaled using the `ScaleData()` function, and principle component analysis (PCA) was carried out. The `FindClusters()` function using the top 30 principal components (PCs) and a resolution of 1.5 was used to determine the initial 29 clusters. Based on the expression of known marker genes we merged clusters that represented the same cell type. Our final list of cell types was: Excitatory neurons, PV Interneurons, SST Interneurons, VIP interneurons, NPY Interneurons, Astrocytes, Vascular-associated cells, Microglia, Oligodendrocytes, and Oligodendrocyte precursor cells.

## Enrichment calculation

Viral vector expression for each of the 861 barcodes across the ten cell types was calculated by averaging the expression of barcoded transcripts across all the individual nuclei that were assigned to that cell type. The relative fold-enrichment in expression toward *Sst+* cells was computed as the ratio of the mean expression in *Sst+* cells and the mean expression in *Sst-* cells: (mean(*Sst+* cells) +0.01)/ (mean(*Sst-* cells)+0.01).

Viral GRE expression for each of the 287 barcodes was calculated at the single-nucleus level as a sum of the expression of the three barcodes that were paired with that GRE. Average GRE-driven expression across the ten cell types was calculated by averaging the expression of the GRE transcripts across all the individual nuclei that were assigned to that cell type. The relative fold-enrichment in GRE expression toward *Sst+* cells was determined as the ratio of the mean expression in *Sst + * cells and the mean expression in *Sst-* cells: (mean(*Sst+* cells)+0.01)/ (mean(*Sst-* cells)+0.01).

## Differential gene expression

To identify which of the GRE-driven transcripts were statistically enriched in *Sst+* vs. *Sst-* cells, we carried out differential gene expression analysis using the R package Monocle2 (*Trapnell et al., 2014*) (*Trapnell et al., 2014*) . The data were modeled and normalized using a negative binomial distribution, consistent with snRNA-seq experiments. The functions `estimateSizeFactors()`, `estimateDispersions()` and `differentialGeneTest()` were used to identify which of the GRE-derived transcripts were statistically enriched in *Sst+* cells. GREs whose false discovery rate (FDR) was less than 0.01 were considered enriched.

## Subsampling GRE reads

A matrix containing counts per cell for GRE12, GRE19, GRE22, GRE44, GRE80 was subsampled using the rbinom function from the 'stats' package in R with the following probabilities (0.5, 0.25, 0.125, 0.0625). The resulting matrix was then analyzed by differential gene expression using the R package Monocle2 as stated above. This process was repeated ten times for each subsampling probability.

## Fluorescence microscopy

### Sample preparation

Mice were sacrificed and perfused with 4% PFA followed by PBS. The brain was dissected out of the skull and post-fixed with 4% PFA for 1–3 days at 4°C. The brain was mounted on the vibratome (Leica VT1000S) and coronally sectioned into 100 μm slices. Sections containing V1 were arrayed on glass slides and mounted using DAPI Fluoromount-G (Southern Biotech).

### Sample imaging

Sections containing V1 were imaged on a Leica SPE confocal microscope using an ACS APO 10x/ 0.30 CS objective (Harvard NeuroDiscovery Center). Tiled V1 cortical areas of ~1.2 mm by ~0.5 mm were imaged at a single optical section to avoid counting the same cell across multiple optical sections. Channels were imaged sequentially to avoid any optical crosstalk.

## Immunostaining

To identify parvalbumin (PV)+ cells, coronal sections were washed three times with PBS containing 0.3% TritonX-100 (PBST) and blocked for 1 hr at room temperature with PBST containing 5% donkey serum. Section were incubated overnight at 4°C with mouse anti-PVALB antibody 1:2000 (Milipore), washed again three times with PBST, and incubated for 1 hr at room temperature with 1:500 donkey anti-mouse 647 secondary antibody (Life Technologies). After washing in PBST and PBS, samples were mounted onto glass slides using DAPI Fluoromount-G.

## Quantification of the percentage of GFP+ cells that were SST+, VIP+, and PV+

Across all images, coordinates were registered for each GFP+ cell that could be visually discerned. An automated ImageJ script was developed to quantify the intensity of each acquired channel for a given GFP+ cell. We created a circular mask (radius = 5.7 μm) at each coordinate representing a

GFP-positive cell, background subtracted (rolling ball, radius = 72 µm) each channel, and quantified the mean signal of the masked area. To identify the threshold intensity used to classify each GFP+ cell as either SST+, VIP+ or PV+, we first determined the background signal in the channel representing SST, VIP or PV by selecting multiple points throughout the area visually identified as background. These background points were masked as small circular areas (radius = 5.7 µm), over which the mean background signal was quantified. The highest mean background signal for SST, VIP and PV was conservatively chosen as the threshold for classifying GFP+ cells as SST+, VIP+ or PV+, respectively.

## Quantification of the distribution of cells as a function of distance from pia

A semiautomated ImageJ algorithm was developed to trace the pia in each image, generate a Euclidean Distance Map (EDM), and calculate the distance from the pia to each GFP+ cell.

## Quantification of the percentage of SST+ cells that were GFP+

An automated algorithm was developed to identify SST+ cells after appropriate background subtraction, image thresholding, masking and filtering for all objects of appropriate size and circularity. The number of SST+ objects (cells) was then counted within a minimal polygonal area that encompassed all GFP+ cells in that image. The ratio of the number of GFP+ cells and SST+ cells within the area of infection (here identified as area with discernable GFP+ cells) was calculated.

## Slice preparation

Acute, coronal brain slices containing visual cortex of 250–300 µm thickness were prepared using a sapphire blade (Delaware Diamond Knives, Wilmington, DE) and a VT1000S vibratome (Leica, Deerfield, IL). Mice were anesthetized though inhalation of isoflurane, then decapitated. The head was immediately immersed in an ice-cold solution containing (in mM): 130 K-gluconate, 15 KCl, 0.05 EGTA, 20 HEPES, and 25 glucose (pH 7.4 with NaOH; Sigma). The brains were quickly dissected and cut in the same ice-cold, gluconate based solution while oxygenated with 95% $O_2$/5% $CO_2$. Slices then recovered at 32°C for 20–30 min in oxygenated artificial cerebrospinal fluid (ACSF) in mM: 125 NaCl, 26 NaHCO3, 1.25 NaH2PO4, 2.5 KCl, 1.0 MgCl2, 2.0 CaCl2, and 25 glucose (Sigma), adjusted to 310–312 mOsm with water.

## Electrophysiological recordings

Using an Olympus BX51WI microscope equipped with a 60x water immersion objective, we used fluorescence illumination to identify rAAV-GRE44-GFP+ (red and green) and rAAV-GRE44-GFP- (only red) SST neurons in the area of injection/AAV infection (*Figure 4a–d*). rAAV-GRE44-GFP- neurons were recorded if they were in the same field of view as rAAV-GRE44-GFP+ neurons under 60x. For rAAV-GRE12-Gq-DREADD-tdTomato experiments (*Figure 4e–j*), tdTomato+ cells and morphologically identified pyramidal neurons in the same field of view under 60x were recorded. Whole-cell current clamp recordings of these neurons in coronal visual cortex slices of P50 to P80 wild-type mice were performed using borosilicate glass pipettes (3–6 MOhms, Sutter Instrument, Novato, CA) filled with an internal solution (in mM): 116 $KMeSO_3$, 6 KCl, 2 NaCl, 0.5 EGTA, 20 HEPES, 4 MgATP, 0.3 NaGTP, 10 $NaPO_4$ creatine (pH 7.25 with KOH; Sigma). Neurobiotin (1.5%) was occasionally included in the internal solution to allow for post-hoc morphological reconstruction of recorded cells. All experiments were performed at room temperature in oxygenated ACSF. Series resistance was compensated by at least 60% in a voltage-clamp configuration before switching to current-clamp ('I Clamp Normal'). After break-in, a systematic series of 1 s current injections ranging from −100 pA to 500 pA were applied to each cell using the User List function in the 'Edit Waveform' tab of pClamp. After such baseline firing rates were calculated, CNO (2 µM, Sigma) was bath applied. An average of at least three trials for each current injection was calculated before and during CNO application.

## Electrophysiological data acquisition and analysis

For electrophysiology, data acquisition of current-clamp experiments was performed using Clampex10.2, an Axopatch 200B amplifier, filtered at 2 kHz and digitized at 20 kHz with a DigiData 1440 data acquisition board (Molecular Devices, Sunnyvale, CA). Analysis of electrophysiological

parameters was done using Clampfit (Molecular Devices, Sunnyvale, CA), Prism 7 (GraphPad Software, La Jolla, CA), Excel (Microsoft, Redmond, WA), and custom software written and generously shared by Dr. Bruce Bean in Igor Pro version 6.1.2.1 (WaveMetrics, Lake Oswego, OR). Membrane potentials in this study were not corrected for the liquid junction potential and are thus positively biased by 8 mV. For analysis of action potential waveform in *Figure 4a–d* and *Supplementary file 4*, the first action potential that appeared during a current injection equivalent to the rheobase was analyzed, as well as the first action potential of the subsequent two current injections. For example, if the rheobase were 20 pA, then all the parameters defined in the next section were also analyzed for the first action potential elicited with 20, 25, and 30 pA of injected current, and averaged.

### Definition of electrophysiological parameters

AP Height (in millivolts): the difference between the peak of the action potential and the most negative voltage during the afterhyperpolarization immediately following the spike.

AP Peak (in millivolts): the most depolarized (positive) potential of the spike.

AP Trough (in millivolts): the most negative voltage reached during the afterhyperpolarization immediately following the spike.

$F_{max}$ initial (in Hertz): the average of the reciprocal of the first three interstimulus intervals, measured at the maximal current step injected before spike inactivation.

$F_{max}$ steady-state (in Hertz): the average of the reciprocal of the last three interstimulus intervals, measured at the maximal current step injected before spike inactivation.

Rate of rise (in volts per second): maximal voltage slope (*dV/dt*) during the upstroke (rising phase) of the action potential.

Rheobase (in picoamperes): the minimal 1000 ms current step (in increments of 5 pA) needed to elicit an action potential.

$R_{in}$ (in megaohms, MΩ): input resistance, determined by using Ohm's law to measure the change in voltage in response to a −50 pA, 1000 ms hyperpolarizing current at rest.

Spike adaptation ratio: the ratio of $F_{max}$ steady-state to $F_{max}$ initial.

Spike width (in milliseconds, used interchangeably with spike half-width): the width at half-maximal spike height as defined above.

$\tau_m$ (in milliseconds): membrane time constant, determined by fitting a monoexponential curve to the voltage chance in response to a −50 pA, 1000 ms hyperpolarizing current at rest.

Threshold (in millivolts): the membrane potential at which dV/dt = 5 V/s.

$V_{rest}$ (in millivolts): resting membrane potential a few minutes after breaking in without any current injection.

## Acknowledgements

We thank D Tom for help with image analysis, Dr. W Renthal and members of the Greenberg lab for discussions, the HMS Single Cell Core for single-nucleus RNA-seq sample processing, Boston Children's Hospital Viral Core for AAV packaging, Dr. B Bean for feedback on electrophysiology, and the Harvard NeuroDiscovery Center Enhanced Neuroimaging Core for imaging and image analysis. This work was supported by the National Institute of Health BRAIN Initiative grant R01 MH114081-01 to MEG, NIH grants MH107620, NS089521, NS108410 to CDH, NIH grant T32GM007753 to MAN, NIH Training in the Molecular Biology of Neurodegeneration grant 5T32AG000222-23 to SH, Vertex Fellowship of the Life Sciences Research Foundation to JG, and Charles A King Trust Postdoctoral Fellowship to HS.

## Additional information

### Funding

| Funder | Grant reference number | Author |
| --- | --- | --- |
| National Institutes of Health | RF1MH11408101 | Michael E Greenberg |
| National Institutes of Health | 5T32AG000222-23 | Sinisa Hrvatin |
| National Institutes of Health | T32GM007753 | M Aurel Nagy |
| National Institutes of Health | MH107620 | Christopher D Harvey |

| National Institutes of Health | NS089521 | Christopher D Harvey |
| National Institutes of Health | NS108410 | Christopher D Harvey |
| Life Sciences Research Foundation | Vertex Fellowship | Jonathan Green |
| Charles A. King Trust | Postdoctoral Fellowship | Hume Stroud |

The funders had no role in study design, data collection and interpretation, or the decision to submit the work for publication.

### Author contributions

Sinisa Hrvatin, Conceptualization, Data curation, Formal analysis, Supervision, Funding acquisition, Validation, Investigation, Visualization, Methodology, Writing—original draft, Project administration, Writing—review and editing; Christopher P Tzeng, Investigation, Visualization, Formal analysis, Writing—original draft, Writing—review and editing; M Aurel Nagy, Investigation, Visualization, Data curation, Formal analysis, Writing—original draft, Writing—review and editing; Hume Stroud, Investigation, Data curation, Formal analysis; Charalampia Koutsioumpa, Oren F Wilcox, Investigation, Validation; Elena G Assad, Investigation; Jonathan Green, Investigation, Writing—review and editing; Christopher D Harvey, Writing—review and editing; Eric C Griffith, Conceptualization, Funding acquisition, Writing—original draft, Writing—review and editing; Michael E Greenberg, Supervision, Conceptualization, Funding acquisition, Writing—original draft, Writing—review and editing

### Author ORCIDs

Sinisa Hrvatin [iD] https://orcid.org/0000-0001-7303-4218
Christopher P Tzeng [iD] https://orcid.org/0000-0003-4734-6298
M Aurel Nagy [iD] https://orcid.org/0000-0003-4608-1152
Hume Stroud [iD] https://orcid.org/0000-0002-2865-5258
Charalampia Koutsioumpa [iD] https://orcid.org/0000-0002-0143-4375
Jonathan Green [iD] https://orcid.org/0000-0003-0434-4225
Eric C Griffith [iD] https://orcid.org/0000-0003-0428-3215
Michael E Greenberg [iD] https://orcid.org/0000-0003-1380-2160

### Ethics

Animal experimentation: This study was performed in strict accordance with the recommendations in the Guide for the Care and Use of Laboratory Animals of the National Institutes of Health. All of the animals were handled according to approved institutional animal care and use committee (IACUC) protocols (#IS00000074-3) of Harvard Medical School. All surgery was performed under isoflurane anesthesia, and every effort was made to minimize suffering.

### Decision letter and Author response

Decision letter https://doi.org/10.7554/eLife.48089.027
Author response https://doi.org/10.7554/eLife.48089.028

## Additional files

### Supplementary files

• Supplementary file 1. Genomic coordinates and conservation scores for the 323,369 reference putative GRE regions.
DOI: https://doi.org/10.7554/eLife.48089.019

• Supplementary file 2. Genomic coordinates, distance to nearest TSS, nearest gene, ATAC-seq-specificity and PESCA-specificity of the 287 GREs that were included in the library.
DOI: https://doi.org/10.7554/eLife.48089.020

• Supplementary file 3. Vector sequences for the in situ evaluated vectors rAAV-ΔGRE/GRE12/GRE22/GRE44-GFP. Gray highlighting indicates the ITRs, green indicates GFP and yellow indicates GRE.
DOI: https://doi.org/10.7554/eLife.48089.021

• Supplementary file 4. Electrophysiological Parameters of GRE44- and GRE44+ SST Neurons in Visual Cortex.
DOI: https://doi.org/10.7554/eLife.48089.022

• Transparent reporting form
DOI: https://doi.org/10.7554/eLife.48089.023

## Data availability

Sequencing data has been deposited in GEO under accession code GSE136802.

The following dataset was generated:

| Author(s) | Year | Dataset title | Dataset URL | Database and Identifier |
|---|---|---|---|---|
| Hrvatin S, Tzeng CP, Nagy MA, Stroud H, Koutsioumpa C, Wilcox OF, Assad EG, Green J, Harvey CD, Griffith EC, Greenberg ME | 2019 | A scalable platform for the development of cell-type-specific viral drivers | http://www.ncbi.nlm.nih.gov/geo/query/acc.cgi?acc=GSE136802 | NCBI Gene Expression Omnibus, GSE136802 |

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
