## [Decision Letter]

Thank you for submitting your article "PESCA: A scalable platform for the development of cell-type-specific viral drivers" for consideration by *eLife*. Your article has been reviewed by three peer reviewers, one of whom is a member of our Board of Reviewing Editors, and the evaluation has been overseen by Catherine Dulac as the Senior Editor. The reviewers have opted to remain anonymous.

The reviewers have discussed the reviews with one another and the Reviewing Editor has drafted this decision to help you prepare a revised submission.

Summary:

This manuscript presents a method to screen sequences for the ability to drive cell type-specific gene expression in the brain. The authors cleverly bring together three recent genomic technologies (cell type-specific ATAC-Seq, single nucleus RNA-Seq, massively parallel reporter assays) to screen for regulatory elements that can drive cell type-specific gene expression. Improved tools to mark specific cell types across mammalian model organisms have the potential to accelerate research by reducing reliance on transgenic mouse models. The technique presented in this manuscript is a substantial improvement over previous methods, which rely upon laboriously screening individual candidates. The proof of principle, a set of gene regulatory sequences which can drive expression in SST neurons, is itself a useful tool that is likely to be used by a number of neuroscience research groups.

Essential revisions:

1) Specificity of the approach: 27% of neurons expressing GFP with the δ-GRE vector are positive for Td-Tomato in the reporter mice, which is much higher than the% of SST neurons in V1. These data suggest the base vector may have some preference for SST neurons on its own. To justify the method as a general strategy to target multiple cell types, the authors should show whether the same base vector can be used in combination with enhancers from other cell types to drive GFP expression in those cells. Another alternative would be to find another base vector that does not have the SST preference without a GRE added and then show that these GREs still restrict expression to SST cells. Finally if the GREs need to work in combination with regulatory elements from the specific vector used here to direct expression to SST cells the results would still be useful, but the readers would need to know this, as it would suggest that the strategy might not work for other cell types with this vector.

2) Sensitivity of the approach: The reviewers raised question about whether the method works "somewhat well", "very well" or is a "broadly useful and scalable platform". To address these questions, more detail is needed about the level of replication across the injected animals – are the data from each animal very similar or very different? Knowing this will be useful for anyone hoping to use this technique. It would be further helpful if the authors could conduct some analysis to determine the viral concentration required to identify a positive vector in the screen. The ideal experiment (spiking in positive control vectors a different concentrations) might take too much time, but reanalysis of their data could be quite valuable (like showing the concentration of all vectors that were positive vs. negative, or where there was disagreement in the three barcodes did it correspond to the library concentration?). Finally, the authors should discuss in more detail, what are the limitations of the current method and how might it be improved? It seems from the text as if the particular single nuclear sequencing method was one weak point due to the very low sensitivity of the method. Could the authors have done as well or better (and proceeded faster) using a more robust sequencing method with pooled nuclei and then screening more candidates? Especially because this is an article for the tools and resources section, a critical and forward-looking Discussion section would make the paper highly valuable to the readership.

3) Subpopulations of SST cells: The reviewers all felt that more data was needed for the authors to conclude that they have labeled a specific subpopulation of SST cells. One possibility is that the virally infected and SST reporter neurons are different from one another because of viral infection. The authors could rule out this possibility experimentally, for example using a low titer non cell-type specific AAV to infect the cortex of SST reporter mice and then compare virally infected to uninfected neurons in the same layers recorded here. Another possibility is that the authors could do more detailed molecular characterization to establish what cell types are being labeled by their vector (snRNA-seq or other method). Through scRNA-seq (Tasic et al.), many of the SST subtypes are now known at high resolution. The authors could use other information (e.g. morphology, marker expression) that is already known about subtypes of SST neurons in addition to layer information to further support their argument that this is a specific subtype of SST cells. Finally we discussed the possibility that the authors could simply drop the claim that this is a subset of SST cells if it cannot be further supported. We do not favor this option as we think the readership of *eLife* will be very interested in new data supporting the existence of subtypes and the utility of these enhancers to target them. However if the authors are unable to provide further data to support the claim, simply removing it from the manuscript would not undermine the publishability of the study.

4) The sequences of the GREs discovered and tested should be reported in the supplementary material of the revised manuscript prior to publication.

---

## [Author Response]

Essential revisions:1) Specificity of the approach: 27% of neurons expressing GFP with the δ-GRE vector are positive for Td-Tomato in the reporter mice, which is much higher than the% of SST neurons in V1. These data suggest the base vector may have some preference for SST neurons on its own. To justify the method as a general strategy to target multiple cell types, the authors should show whether the same base vector can be used in combination with enhancers from other cell types to drive GFP expression in those cells. Another alternative would be to find another base vector that does not have the SST preference without a GRE added and then show that these GREs still restrict expression to SST cells. Finally if the GREs need to work in combination with regulatory elements from the specific vector used here to direct expression to SST cells the results would still be useful, but the readers would need to know this, as it would suggest that the strategy might not work for other cell types with this vector.

We agree with the reviewers that the vector without the enhancer appears to show preference toward SST+ cells. To address whether the same vector can be used to target other cell types, we took the reviewer’s experimental suggestion and cloned the mDlx5/6 enhancer (Dimidschstein et al., 2016) into the PESCA vector, and showed that the resulting vector expression pattern resembles the previously reported one. We reported this experiment as follows: “To show that not all the enhancers exhibited strong SST-specificity using our viral vector, we cloned the mDlx5/6 enhancer whose expression was restricted to a broader population of inhibitory neurons. We injected the rAAV2/9-mDlx5/6-GFP vector and observed that 42.9% of GFP^+^ cells were also positive for tdTomato (1977 cells, 3 animals, Figure 3—figure supplement 2).”

Since we do not, at this time, have access to other cell-type- or cell-class-specific enhancers, we wanted to highlight a recent bioRxiv preprint (Graybuck et al., 2019) which reports combining enhancers with the minimal β-globin promoter to achieve cell-type-specific expression in cortical L5 excitatory neurons. This study appears to use the same promoter as our PESCA construct, suggesting that our PESCA construct could likely be used to target other neuronal cell types.

We agree with the reviewers that the AAV tropism and perhaps the choice of the promoter affects the pattern of expression. In an extreme case, if an AAV serotype does not infect the cell type of interest, a PESCA screen will likely fail to identify GREs capable of driving expression in that cell type.

We added a Discussion section to the paper highlighting these considerations and noting that while our base AAV9 vector appears to have a broad pattern of expression across the V1 cortex, it may not be appropriate for all PESCA screens. While these considerations of viral tropism and choice of promoter are not unique to our approach and apply to all viral tools, the challenges of generating a cell-type-specific viral vector for any cell type in the body, highlights, in our view, the value of an approach such as PESCA, which can be extended to screen through combinations of enhancers, promoters as well as viral serotypes to identify the optimal cell-type-specific vector. To this end, we’ve made our PESCA library cloning strategy largely vector-independent and modular such that different promoters can be tested if needed by changing the final cloning step, or different serotypes by repackaging the final library. We have highlighted these considerations in the discussion: “Indeed, the PESCA library cloning strategy is largely vector- and capsid-independent, allowing for the use of different promoters or serotypes. Our choice of capsid and promoter was driven by previous work using AAV9 and the minimal beta-globin promoter to drive expression in cortical interneurons, however, different capsids or promoter may be preferred for targeting other cell types.”

2) Sensitivity of the approach: The reviewers raised question about whether the method works "somewhat well", "very well" or is a "broadly useful and scalable platform". To address these questions, more detail is needed about the level of replication across the injected animals – are the data from each animal very similar or very different? Knowing this will be useful for anyone hoping to use this technique. It would be further helpful if the authors could conduct some analysis to determine the viral concentration required to identify a positive vector in the screen. The ideal experiment (spiking in positive control vectors a different concentrations) might take too much time, but reanalysis of their data could be quite valuable (like showing the concentration of all vectors that were positive vs. negative, or where there was disagreement in the three barcodes did it correspond to the library concentration?).

We agree with the reviewers that these data would be a very valuable addition to the manuscript. We have now included data regarding the correlation in enrichment scores across animals: “As expected, a high correlation was observed between GRE-specific enrichment scores across two animals (r = 0.54, p < 2.2×10^-16^) (Figure 2—figure supplement 5B)”.

We further analyzed our data and now report and discuss the relationship between the number of GRE transcripts detected and the measured specificity: “To assess how the abundance of each GRE in the library impacts our ability to detect cell-type-specific expression, we analyzed the specificity of each GRE as a function of the number of transcripts that we retrieved. We observed that highly abundant GRE-driven transcripts were more likely to be significantly enriched in SST^+^ cells suggesting that we may not have had sufficient power to assess the cell-type-specificity of the less abundant GREs in the library. (Figure 2—figure supplement 5F).

We also highlight this point further as a part of our Discussion: “…This is particularly relevant insofar as the ability to assess the GRE library depends on the number of cells sequenced from the target and non-target populations and the sequencing depth. In the screen described here, we estimate having sufficient power to assess approximately 2/3 of the 287 GREs at the reported sequencing depth (Figure 2—figure supplement 5).”

We also performed an analysis in which we computationally subsampled the counts from our top 5 viral hits to determine the sequencing depth at which we lose the ability to detect them as significantly enriched in SST+ cells: “…Consistent with this observation, computationally subsampling the number viral transcripts across our most cell-type-specific GREs gradually reduced our ability to statistically detect their enrichment in Sst^+^ cells (Figure 2—figure supplement 6). These observations suggest that the expression of sparsely detected GRE-driven transcripts may not be sufficient to allow evaluation of cell-type-specificity and that by increasing sequencing depth we may be able to screen and evaluate of a larger number of GREs.”

Finally, the authors should discuss in more detail, what are the limitations of the current method and how might it be improved? It seems from the text as if the particular single nuclear sequencing method was one weak point due to the very low sensitivity of the method. Could the authors have done as well or better (and proceeded faster) using a more robust sequencing method with pooled nuclei and then screening more candidates? Especially because this is an article for the tools and resources section, a critical and forward-looking Discussion section would make the paper highly valuable to the readership.

We thank the reviewers for the suggestion and have added the following discussion to the manuscript: “In the future, several factors should be considered to facilitate the further optimization of the PESCA methodology for the development of cell-type-specific vectors. […] In this regard, the snRNA-seq PESCA screen identified GRE12, GRE22 and GRE44 as, respectively, 8.3-, 9.1- and 7.2-fold more highly expressed in SST^+^ compared to SST^-^ cells, whereas these GREs showed distinct specificity for SST^+^ cells (91%, 73% and 96% respectively, Figure 3F) when evaluated at the protein level, a finding which could be attributed to a variety of factors.”

3) Subpopulations of SST cells: The reviewers all felt that more data was needed for the authors to conclude that they have labeled a specific subpopulation of SST cells. One possibility is that the virally infected and SST reporter neurons are different from one another because of viral infection. The authors could rule out this possibility experimentally, for example using a low titer non cell-type specific AAV to infect the cortex of SST reporter mice and then compare virally infected to uninfected neurons in the same layers recorded here. Another possibility is that the authors could do more detailed molecular characterization to establish what cell types are being labeled by their vector (snRNA-seq or other method). Through scRNA-seq (Tasic et al.), many of the SST subtypes are now known at high resolution. The authors could use other information (e.g. morphology, marker expression) that is already known about subtypes of SST neurons in addition to layer information to further support their argument that this is a specific subtype of SST cells. Finally we discussed the possibility that the authors could simply drop the claim that this is a subset of SST cells if it cannot be further supported. We do not favor this option as we think the readership of eLife will be very interested in new data supporting the existence of subtypes and the utility of these enhancers to target them. However if the authors are unable to provide further data to support the claim, simply removing it from the manuscript would not undermine the publishability of the study.

Per reviewers’ suggestion we carried out in situ hybridization to identify whether GRE44 expression is enriched in any of the four major molecularly-defined SST^+^ subsets (CrH^+^, Nos1^+^, Hpse^+^ or Nr2f2^+^ cells)(Tasic et al., 2018). To detect the expression of the viral RNA, we used probes against the Gfp and the Wpre sequence. During our analysis, we observed two distinct patters of Wpre staining: A largely cytoplasmic signal which overlapped with many Sst^+^ cells and a punctate nuclear signal which appeared in a broader set of cells. The latter signal may be due to the in situ probe binding to the viral genome, the viral RNA, or both. Because this punctate signal appears in a broad set of cells including Sst^-^ cells (and therefore does not correspond well to the distribution of GFP protein expression), we are hesitant to further pursue the RNAscope approach to analyze the specificity of GRE44 expression before we fully understand the origin of these signals. Since the reviewers’ discussed these experiments as optional, we have instead removed claims that the GRE44 virus drives cell-type-specific expression in a discrete subpopulation of somatostatin interneurons. For example, we removed: “and raising the possibility that these GREs may restrict expression to a specific subtype(s) of SST^+^ interneurons” and we clarified: “Although we cannot confirm that GRE44 expression is restricted to an individual subtype of SST interneurons, our electrophysiology experiments further emphasize the potential of PESCA to target functionally distinct subgroups of previously defined interneuron types.”

4) The sequences of the GREs discovered and tested should be reported in the supplementary material of the revised manuscript prior to publication.

The genomic coordinates for all the GREs tested and the full sequences of the GRE12/22/44-GFP vectors are now reported in Supplementary file 3.